# Detection of Two Highly Diverse Peribunyaviruses in Mosquitoes from Palenque, Mexico

**DOI:** 10.3390/v11090832

**Published:** 2019-09-07

**Authors:** Anne Kopp, Alexandra Hübner, Florian Zirkel, Daniel Hobelsberger, Alejandro Estrada, Ingo Jordan, Thomas R. Gillespie, Christian Drosten, Sandra Junglen

**Affiliations:** 1Berlin Institute of Health, Institute of Virology, Charité—Universitätsmedizin Berlin, corporate member of Freie Universität Berlin, Humboldt-Universität zu Berlin, 10117 Berlin, Germany (A.K.) (A.H.) (C.D.); 2Institute of Virology, University of Bonn Medical Centre, Bonn, Germany, Biotest AG, 63303 Dreieich, Germany; 3Department of Biology, Universität Regensburg, 93053 Regensburg, Germany; 4Estación de Biología Tropical Los Tuxtlas, Instituto de Biología, Universidad Nacional Autónoma de México, Mexico City 04513, Mexico; 5ProBioGen AG, 13086 Berlin, Germany; 6Department of Environmental Sciences and Program in Population Biology, Ecology and Evolution, Emory University, Atlanta, GA 30322, USA; 7Department of Environmental Health, Rollins School of Public Health, Emory University, Atlanta, GA 30322, USA

**Keywords:** *Peribunyaviridae*, orthobunyavirus, arbovirus, insect-specific virus, mosquito, Mexico

## Abstract

The *Peribunyaviridae* family contains the genera *Orthobunyavirus*, *Herbevirus*, *Pacuvirus*, and *Shangavirus*. Orthobunyaviruses and pacuviruses are mainly transmitted by blood-feeding insects and infect a variety of vertebrates whereas herbeviruses and shangaviruses have a host range restricted to insects. Here, we tested mosquitoes from a tropical rainforest in Mexico for infections with peribunyaviruses. We identified and characterized two previously unknown viruses, designated Baakal virus (BKAV) and Lakamha virus (LAKV). Sequencing and de novo assembly of the entire BKAV and LAKV genomes revealed that BKAV is an orthobunyavirus and LAKV is likely to belong to a new genus. LAKV was almost equidistant to the established peribunyavirus genera and branched as a deep rooting solitary lineage basal to herbeviruses. Virus isolation attempts of LAKV failed. BKAV is most closely related to the bird-associated orthobunyaviruses Koongol virus and Gamboa virus. BKAV was successfully isolated in mosquito cells but did not replicate in common mammalian cells from various species and organs. Also cells derived from chicken were not susceptible. Interestingly, BKAV can infect cells derived from a duck species that is endemic in the region where the BKAV-positive mosquito was collected. These results suggest a narrow host specificity and maintenance in a mosquito–bird transmission cycle.

## 1. Introduction

The family *Peribunyaviridae* (order *Bunyavirales*) comprises the genera *Orthobunyavirus, Herbevirus, Pacuvirus* and *Shangavirus*. The genus *Orthobunyavirus* contains 88 species and 15 serogroups according to the current ICTV report [1]. Orthobunyaviruses circulate between blood-feeding arthropods and vertebrates. They can infect a wide variety of different hosts including birds, livestock, and humans. Clinical manifestations can range from acute but self-limiting fever, joint pain or rash to more severe symptoms like encephalitis or haemorrhagic fever [2,3]. The viruses show an almost worldwide distribution. Important orthobunyaviruses occurring in Central and South America are, for example, Oropouche virus of the Simbu serogroup, and Cache Valley virus and Kairi virus of the Bunyamwera serogroup [4,5]. Orthobunyaviruses have a tripartite negative-sense RNA genome with consensus terminal nucleotides at the ends of each genome segment. The large segment (L segment) encodes for an RNA-dependent RNA polymerase (RdRp). The medium segment (M segment) encodes for the glycoproteins Gn and Gc as well as for the non-structural protein NSm. The small (S segment) encodes two proteins in an overlapping open reading frame (ORF), the nucleocapsid protein (N) and the non-structural protein (NSs), which are both translated from the same mRNA using alternative start codons [6,7].

The genus *Pacuvirus* includes three species, Pacui virus (PACV), Rio Preto da Eva virus (RPEV) and Tapirape virus (TPPV) which were isolated either from rodents or sand-flies collected in Brazil or Trinidad [8,9]. Although Pacuviruses seem not to code for a NSs protein, which is an important interferon antagonist, they are able to infect vertebrates.

Herbeviruses have the simplest genome organization within the family *Peribunyaviridae*, as they lack the non-structural proteins NSm and NSs [10]. The type species Herbert virus (HEBV) was initially isolated from *Culex* mosquitoes sampled in Côte d’Ivoire and was subsequently detected in *Culex* sp. mosquitoes from Ghana [10]. Two other herbeviruses, Tai virus (TAIV) and Kibale virus (KIBV) were isolated from *Culex* sp. mosquitoes from Côte d’Ivoire and Uganda, respectively. Herbeviruses are unable to infect vertebrates and are so called insect-specific viruses [10,11,12,13]. 

*Insect shangavirus* (former *Shuangao Insect virus 1*) is the only species of the genus *Shangavirus*. It was sequenced from a pool of mixed insects (*Chrysopidae* and *Psychoda alternata*) and it is believed to be insect-specific [14]. No virus isolate is available. The predicted genome organization of shangaviruses is comparable to that of orthobunyaviruses and herbeviruses, e.g., like herbeviruses they seem to lack a NSs while they seem to encode an NSm as typical for orthobunyaviruses [14].

Khurdun virus (KHURV) defines an isolated lineage that shares a most recent common ancestor with the clade comprising orthobunyaviruses and pacuviruses [15,16]. KHURV was isolated from coot in the Volga River delta but so far no other vertebrate host or insect vector has been identified.

Several arboviruses are emerging in the Americas. They have either been introduced from other continents, e.g., Zika virus and Chikungunya virus, or endemic viruses are spreading to new geographic regions mainly due to anthropogenic land-use changes. For example, the alphavirus Mayaro virus (MAYV, family *Togaviridae*) which is endemic in the tropical rainforests of South America and the Caribbean basin, has been spreading into densely populated parts of American countries and causing sporadic outbreaks [17,18]. Mosquitoes of the genus *Hemagogus* mainly transmit the MAYV. However, laboratory experiments identified *Aedes aegypti* and *Aedes albopictus* mosquitoes as potent MAYV vectors promoting fear that MAYV may establish an urban transmission cycle [17,19]. Another example is the increase of infections with the orthobunyavirus Oropouche virus (OROV), which can be linked to human invasion into forest habitats [5,20,21,22]. In addition to already known viruses, previously unknown arboviruses are also emerging, highlighting the importance for surveillance studies in rural tropical regions. For example, a novel orthobunyavirus, named Itaya virus, was isolated from febrile patients from the Amazon basin in Peru [23]. 

Here we sought to assess the genetic diversity of peribunyaviruses in Neotropical mosquitoes collected in the primary lowland rainforest of the area of the Palenque National Park, southeastern Mexico.

## 2. Materials and Methods

### 2.1. Mosquito Sampling and RT-PCR Screening

Adult mosquitoes were collected in the area of the Palenque National Park in southeastern Mexico between July and September 2008 using BG-sentinel traps (Biogents, Regensburg, Germany). Mosquito identification, RNA extraction, and cDNA synthesis was performed as described previously [24]. Mosquitoes were tested in pools of ten specimens for peribunyaviruses by a generic reverse transcription (RT) PCR targeting the RdRp gene using Platinum^®^
*Taq* polymerase (Life Technologies, Darmstadt, Germany). The first round PCR mixture (25 µL) contained 2 µL cDNA as template, 1× buffer, 2.5 mM MgCl_2_, 0.2 mM dNTPs, 0.6 µM forward and reverse primer, 0.1 µL Platinum Taq polymerase. Components and concentrations of the hemi-nested PCR mixture were similar to the mixture described above, but 0.5 µL of the first round PCR product served as template. First round PCR was carried out with the primers Peri-F1 5′-CAAARAACAGCAAAAGAYAGRGARA and Peri-R1 5′-TTCAAATTCCCYTGIARCCARTT, followed by a hemi-nested PCR with Peri-F2 5′-ATGATTAGYAGRCCDGGHGA and Peri-R1, respectively. The thermal cycling protocol included the following steps: 3 min at 95 °C, ten touch down cycles of 15 s at 95 °C, 20 s at 55 °C (−0.5 °C per cycle), 40 s at 72 °C, 35 cycles of 15 s at 95 °C, 20 s at 50 °C, 40 s at 72 °C and a final elongation step at 72 °C for 5 min. Initial sequence fragments were elongated using fragment specific primers and generic primers based on conserved regions of the RdRp gene of closely related viruses using Platinum^®^
*Taq* polymerase (Life Technologies, Darmstadt, Germany). The elongated sequences cover the third conserved region of the RdRp, containing motifs Premotif A and motifs A to E, and were used for comparison with the NCBI database.

To identify virus-positive mosquitoes of virus-positive pools, RNA was extracted from individual mosquitoes using the RNeasy Kit (Qiagen, Hilden, Germany) and cDNA was synthesized using random hexamer primers and Superscript III (Life Technologies, Darmstadt, Germany). Samples were tested by PCR with virus specific primers. For confirmation of the mosquito species of virus positive individuals, the cytochrome c oxidase 1 gene was sequenced [24,25].

### 2.2. Genome Sequencing

The full genomes of BKAV and LAKV were obtained by a combination of Illumina Next generation sequencing (NGS) and conventional RT-PCR. For BKAV the supernatant of infected C6/36 cells was purified by ultracentrifugation through a 36% sucrose cushion followed by an ultracentrifugation through a continuous sucrose gradient. Virus concentration in the fractions was measured by quantitative real-time PCR (qRT-PCR) and virus particles of the fractions with the highest virus concentrations were pelleted by ultracentrifugation. RNA extraction was performed with the RNeasy Kit (Qiagen, Hilden, Germany) according to the manufacturer’s instructions. cDNA synthesis was done with Maxima H Minus Double-Stranded cDNA Synthesis Kit (Thermo Fisher Scientific, Waltham, MA, USA) and random hexamer primers. For LAKV RNA was extracted from the individual virus-positive mosquito as described above and transcribed into cDNA using the Superscript One-Cycle cDNA Kit (Life Technologies, Darmstadt, Germany). For both viruses, DNA libraries were prepared using the Nextera XT DNA Sample Preparation Kit (Illumina, San Diego, CA, USA). Sequencing was performed on the MiSeq desktop sequencer with the MiSeq Reagent Kit v3 (Illumina, San Diego, CA, USA). Genome gaps were amplified by conventional PCR with sequence specific primers using Platinum^®^
*Taq* polymerase (Life Technologies, Darmstadt, Germany) and Sanger sequenced. Genome ends of the BKAV and the LAKV were determined using the 5′RACE System for Rapid Amplification of cDNA Ends (Life Technologies, Darmstadt, Germany). 

### 2.3. Genome and Phylogenetic Analyses

Identified genome segments were analyzed for open reading frames and conserved motifs using Geneious R9. Pfam (http://pfam.xfam.org/) and SignalP 4.1 (http://www.cbs.dtu.dk/services/SignalP/) were used to identify signal peptide sequences. Potential glycosylation sites were identified using the NetNGlyc 1.0 server (http://www.cbs.dtu.dk/services/NetNGlyc/). Prediction of putative transmembrane domains was performed with the TMHMM tool (http://www.cbs.dtu.dk/services/TMHMM/).

For phylogenetic analyses, deduced amino acid sequences of each genome segment were aligned with sequences of representative peribunyaviruses using the MAFFT (E-INS-i, Blosum62) alignment tool implemented in Geneious. Maximum likelihood trees were inferred using the Blosum62 substitution model. Bootstrap analyses were performed using 1000 replicates. Tomato spotted wilt virus (TSWV), the type species of the genus *Orthotospovirus* (family *Tospoviridae*), was used as an outgroup.

### 2.4. Cell Culture and Virus Replication Analyses

Supernatants of homogenized mosquitoes were used to inoculate subconfluent *Aedes albopictus* (C6/36) and African green monkey kidney (VeroE6) cells. C6/36 cells were incubated at 28 °C without CO_2_ and VeroE6 cells at 37 °C and 5% CO_2_, respectively. An aliquot of the supernatant was either passaged onto fresh cells when cytopathic effects (CPE) appeared or seven days post infection (dpi). Cell culture supernatants were investigated for virus presence using virus-specific RT-PCR. RNA was extracted from cell culture supernatant using the Macherey-Nagel Nucleospin Viral RNA kit according to the manufacturer’s instructions. cDNA synthesis was performed using the SuperScript III RT System (Life Technologies GmbH, Darmstadt, Germany) and random hexamer primers. Numbers of infectious particles were determined by TCID_50_ titration [10]. Replication kinetics were performed on the following insect and vertebrate cell lines: C6/36 and U4.4 (*Aedes albopictus* larvae), VeroE6 (African green monkey, kidney), BHK-J and BHK-21 (Baby hamster, kidney), S.hisp. (cotton rat, *Sigmodon hispidus*, lung [24]), Tb1Lu (bat, *Tadarida brasiliensis*, lung [24]), Huh7 (human liver carcinoma cell line), A549 (human lung carcinoma cell line), HEK293T (human embryonic kidney cells), U373 (human glioblastoma cell line), AGE1.CR (muscovy duck, primary embryo cells [26]), DF1 (chicken fibroblasts [27]), and LMH (chicken hepatocellular carcinoma [28]). Cells were infected with multiplicities of infection (MOI) ranging from 0.01–10. Supernatants of the vertebrate cell lines VeroE6, BHK-J, BHK-21, Tb1Lu, S.hisp, Huh7, A549, HEK293T, and U373 infected with MOI 10 were passaged two additional times after 7 days post infection. Blind cell culture passages with 100 µL cell culture supernatant of the previous cell culture passage mixed with 200 µL media without additives were used for inoculation of fresh cells. Viral genome copies in cell culture supernatants were determined using qRT-PCR using the following set of probe and primers: BKAV-real-probe 5′-TCTCCGCCAGTGGGACTTGACTCTCTT, BKAV-real-F 5′-GCCCGTTCAAATTCAAGCAT, BKAV-real-R 5′-TGTCATTTCAGCATCCAATACGA. Temperature sensitivity assays were performed with the BKAV, HEBV, or LACV, as described previously [12]. Briefly, C6/36 cells were infected in duplicates with MOI 0.01 with the BKAV, HEBV, or LACV, respectively. Cells where incubated at three different temperatures: 28, 31, and 34 °C. Viral genome copies in cell culture supernatants were determined for three consecutive days using qRT-PCR.

### 2.5. Nucleotide Sequence Accession Numbers

The full genomic sequences of BKAV and LAKV were deposited in GenBank under accession numbers MN092352–MN092354 and MN092355–MN092357, respectively. The WYOV–Palenque sequence has the GenBank accession number MN092358.

## 3. Results

### 3.1. Identification of Orthobunyaviruses and Herbeviruses

Three peribunyavirus positive mosquito pools (0.8%; 3/371) were identified by generic RT-PCR. The conserved RdRp motifs, Premotif A and motifs A to E, were amplified from virus positive pools using fragment specific and generic primers based on conserved regions of the RdRp gene. Comparison of the three sequences to the NCBI database revealed identities of 74%, 87%, and 68% to the orthobunyaviruses—Koongol virus (KOOV) and Wyeomyia virus (WYOV)—as well as to insect-specific viruses of the genus *Herbevirus*, respectively, suggesting the identification of two novel species and one novel strain, designated as Baakal virus (BKAV), Lakamha virus (LAKV) and Wyeomyia virus strain Palenque (WYOV-Palenque) (Table 1). The translated amino acid sequence of WYOV–Palenque showed 96.8% aa identity to the WYOV strain Darien, indicating the first detection of WYOV in Mexico. WYOV–Palenque was named after the Palenque National Park from where the mosquitoes originated. BKAV and LAKV were named with regard to the history of the area. Baakal means bone in Maya and is the historical kingdom around the city of Palenque. Lakamha is the historical name of Palenque given by the Maya and means Big Water. Testing of individual mosquitoes of the three virus-positive pools showed that only one single mosquito was positive in each case (0.03%; 1/3491). Morphological mosquito species identification was confirmed by amplification of cytochrome *c* oxidase I genes and sequence comparison to BOLD systems databases. BKAV was detected in a *Culex nigripalpus* mosquito and LAKV was found in a *Wyeomyia complosa* mosquito. The species of the WYOV–Palenque positive mosquito could neither be identified morphologically nor by the COI gene sequence. The COI gene sequence showed 93.1% similarity to *Wyeomyia complosa* indicating that the mosquito is a member of the Tribe *Sabethini* to which the genera *Wyeomyia*, *Limatus*, *Trichoprosopon* and *Sabethes* belong. 

### 3.2. Full Genome Sequencing and Phylogenetic Relationship

The full genomes of BKAV and LAKV were sequenced by NGS either from cell culture supernatant or from the individual virus-positive mosquito, as isolation attempts of LAKV were not successful (see below). Although infectious cell culture supernatant was used for BKAV, only 59 virus specific reads were obtained which were distributed over the entire genome, resulting in a low coverage of 29%, 47%, and 26% for the L, M, and S segments, respectively (Figure 1A, Table 2). In contrast, almost the entire LAKV genome was *de novo* assembled directly from the single mosquito (Figure 1B, Table 2). Sequence gaps were amplified using conventional RT-PCR and fragment specific primers. Genome ends were amplified by RACE-PCR. Pairwise amino acid identities for BKAV and LAKV RdRp and N proteins compared to those of representative peribunyaviruses are given in Table 3. Maximum likelihood analyses based on all genome segments showed that the WYOV–Palenque grouped with WYOV and that BKAV branched as a sister species to KOOV (Figure 2). LAKV shares a most recent common ancestor (MRCA) with herbeviruses and defines a new deep rooting lineage (Figure 2). 

### 3.3. Genome Analyses

The full genome of BKAV showed the typical genome organization of orthobunyaviruses including the three segments L, M and S and the conserved terminal nucleotides. The glycoprotein precursor (GPC), encoded on the M segment, showed the conserved protein cleavage sites and is predicted to be cleaved into Gn, NSm, and Gc proteins (Figure 1A and Figure 3). Interestingly, the 3′ and 5′ UTRs of the BKAV M segment are of similar length (272 nt and 265 nt, respectively) and are much longer than the UTRs of other orthobunyaviruses (e.g., BUNV 3′UTR is 56 nt and the 5′ UTR is 100 nt). Unfortunately, no information on KOOV and UMBV UTRs is available. BKAV contains the conserved amino acids of the Gn zinc finger motif and Gc fusion peptide suggesting similar protein functions (Figure 4). Two putative CTG start codons in a Kozak consensus sequence were found for the NSs protein either generating a protein of 88 aa or of 97 aa. 

The LAKV genome was different from that of established peribunyaviruses. Neither NSm nor NSs genes were found for LAKV, as also observed for herbeviruses. However, all LAKV genome segments contained the conserved terminal nucleotides of orthobunyaviruses (Figure 3). The LAKV GPC is significantly shorter than that of orthobunyaviruses and ca. 20 aa shorter than that of herbeviruses (Figure 4A) [10]. Interestingly, LAKV and herbeviruses lack the variable region of the Gc protein. Putative Gn and Gc signal peptide cleavage sites were identified. The putative predicted LAKV Gn C-terminus (IIS_298_) did not show the highly conserved arginine present in orthobunya- and pacuviruses and also differed from herbeviruses. A Gn zinc finger motif containing conserved amino acids, e.g., cysteine residues, was also identified (Figure 4B). The Gc fusion protein was found to be more distinct and shared less conserved amino acids with other peribunyaviruses. However, the conserved GxC sequence motif in the C-terminal region of the Gc fusion motif was present (Figure 4C). The observed differences in the genome organization of LAKV compared to those of ortho- and herbeviruses are in agreement with its phylogenetic placement on a long solitary branch in equidistance to the four established genera.

### 3.4. Virus Isolation and Growth Analyses in Insect and Vertebrate Cells

BKAV induced weak CPE after four days post infection (dpi) on C6/36 cells. Virus isolation attempts of WYOV–Palenque and LAKV, and of BKAV on VeroE6 cells failed. To further analyze growth characteristics of BKAV and to get insight into its putative host range, different mosquito and vertebrate cell lines were infected. BKAV replicated up to 100-fold better on immune competent mosquito cells (U4.4) than on DICER deficient mosquito cells (C6/36) at 1 and 2 dpi (Figure 5A). However, BKAV reached a similar plateau of >10^9^ RNA genome copies/mL 4 dpi on both cell lines. Surprisingly, BAKV did not replicate on any of the tested vertebrate cells infected with an MOI 10, comprising human (Huh7), non-human primate (VeroE6), hamster (BHK-21 and BHK-J), cotton rat (S. hisp), and bat (Tb1Lu) cells (Figure 5B). Other orthobunyaviruses, including the closest relatives of BKAV (KOOV, UMBV and GAMV) replicate efficiently on vertebrate cells [29] suggesting that BKAV differs from orthobunyaviruses isolated to date. We next tested if the replication of BKAV is sensitive to vertebrate body temperatures, which may point towards a host range restriction to insects [12]. We performed temperature sensitivity assays on C6/36 cells using BKAV, as well as the insect-specific HEBV and the classical arbovirus La Crosse virus (LACV) as controls. As expected, replication of LACV increased with higher temperatures (Figure 5D), whereas the replication of HEBV was impaired at 31 °C and blocked at 34 °C (Figure 5E). BKAV replication was 10- to 100-fold more efficient at temperatures of 31 and 34 °C compared to lower temperature of 28 °C (Figure 5C) suggesting that BKAV is not an insect-specific virus. To further investigate the putative host range of BAKV, we examined the vertebrate hosts of the closest relatives of BAKV. KOOV, UMBV and GAMV seem to be maintained in a mosquito–bird transmission cycle [29,30]. Thus, we infected the two chicken cell lines LMH and DF1, commonly used in influenza research and a cell line derived from a Muscovy duck (*Cairina moschata,* AGE1.CR) [26,31], which is a species native to Mexico, Central and South America. The BKAV was not able to replicate in cells derived from chicken (Figure 5F) but surprisingly replicated in the Muscovy duck cells (Figure 5G), suggesting that BKAV is capable of maintenance in a mosquito–bird transmission cycle. 

To this end it is unclear if the susceptibility of cells derived from Muscovy ducks for BKAV points towards host or cell type specificity or indicates a specific sensitivity for the type of antiviral response. We thus infected a repertoire of different cell lines to investigate this further. AGE1.CR cells express the adenovirus early antigen protein 1A that inhibits cellular antiviral responses [26,32]. In addition, ducks express RIG-I, which senses cytoplasmic RNA and induces the expression of interferon-stimulated genes, whereas RIG-I seems to be absent in chicken [33]. To test if BAKV relies on E1A mediated protection and is able to overcome the RIG-I mediated innate immune response, we infected HEK293T cells, which express E1A and are used to propagate interferon sensitive viruses [32,34,35], as well as A549 cells, which do not express E1A but RIG-I. Both cell lines were not susceptible for BKAV suggesting that neither E1A nor RIG-I determine BKAV infection (Figure 5B). As AGE1.CR cells were derived from neuronal cells, we further examined if BKAV has a neurotropism by infecting a human glioblastoma cell line (U373). BKAV was also not able to replicate in U373 cells (Figure 5B). In summary, the data on host and cell tropism suggest that the BKAV is highly host-specific. 

## 4. Discussion

In this study we identified two novel viruses, BKAV and LAKV, and detected WYOV for the first time in Mexico. The phenotypic characterization of BKAV provided evidence that BKAV is a host-specific arbovirus, which can neither infect cell lines derived from mammalian species commonly used in virus research nor avian cell lines that are widely susceptible for viral infection. Only cells derived from a duck species occurring in the region where BKAV was found in mosquitoes were susceptible. Further in depth studies are needed to identify host-specificity factors, such as the presence of a receptor for viral attachment and entry, as well as to characterize the role of the BKAV NSs protein. Of note, growth curve analyses of previously unknown viruses over an increasing temperature range in mosquito cells can be used as surrogates for risk assessment to differentiate between mosquito-borne and mosquito-specific viruses when no susceptible vertebrate cell line can be identified [12]. 

BKAV was isolated from *Culex nigripalpus* mosquitoes, a species known to feed primarily on birds [36]. The closest relatives of BKAV are viruses of the Koongol (Koongol virus), Turlock (Umbre virus) and Gamboa (Gamboa virus) serogroups, which also seem to be maintained in a mosquito–bird transmission cycle. KOOV was isolated in Australia and New Guinea from *Culex* and *Ficalbia* mosquitoes and antibodies were detected in mammals, marsupials, and birds [37,38,39]. Umbre virus was detected in *Culex* mosquitoes and birds from India and Malaysia [40] and anti-Umbre virus antibodies were found in bird sera from Malaysia [41]. Other viruses of the Turlock serogroup (the Lednice and Turlock viruses) seem to be distributed in the Americas, Africa, and Europe, but no sequence information of these viruses is available [42]. Viruses of the Gamboa serogroup were only detected in tropical regions of Central and South America and are transmitted between *Aedes squamipennis* mosquitoes and birds [29,43,44,45]. In summary, the data on BAKV susceptible cell lines, feeding preferences of mosquito vector species, as well as the maintenance cycles of related viruses suggest that BKAV is maintained in a mosquito–bird transmission cycle. Future investigations of sera from wild birds of the Palenque region could potentially identify which bird species can be infected by BKAV. As no disease in humans, domestic or wild animals is associated with BKAV-related viruses and as only one specific avian cell line supports replication of BKAV, the novel virus is not expected to be able to cause disease in humans or livestock. 

The BKAV N protein showed a maximum identity of 63% to the KOOV N protein suggesting that BKAV belongs to a previously unknown serogroup, as N proteins of the same serogroup are more than 80% identical [46,47]. Differences were found in the predicted sizes of NSm and NSs proteins of BKAV and related viruses. The NSm protein is involved in virus assembly and budding and is usually conserved in orthobunyaviruses [3]. The NSm of the KOOV is truncated (9 kDA) whereas that of GAMV is elongated (36 kDA) when compared to that of BKAV (20 kDA) [30,45]. The NSm protein of BUNV was shown to be important for virus assembly and morphogenesis. However, a recent study using recombinant Oropouche virus showed that virus replication in cell culture was not affected by an NSm deletion [48]. Similar observations were made for the Maguari virus [49]. The NSs protein acts as an interferon antagonist and is involved in host protein shutoff [3,50,51]. The NSs ORF of BKAV contains two putative start codons one upstream and one downstream of the N ORF start codon. This was recently described for Enseada virus (family *Peribunyaviridae*, genus *Orthobunyavirus*) as well [52]. Both BKAV start codons exhibit a CTG instead of a classical ATG and are in a weak Kozak sequence context. Alternate start codons and a weak Kozak context could reduce gene expression [53,54]. However, if this is true for avian viruses needs to be studied. Further studies investigating the role of bird-associated NSm and NSs proteins including those of BKAV may help to identify specifics compared to mammalian orthobunyaviruses.

Orthobunyavirus UTRs have important functions in the viral life cycle, e.g., they contain promoter elements for transcription and translation [6,55,56]. BKAV showed the characteristic conserved terminal nucleotides at each genome segment but interestingly had longer M segment 3′- and 5′UTRs than other orthobunyaviruses. Differences of orthobunyavirus UTRs within segments of one virus or between related viruses have been reported previously [57]. Transcription and translation of each genome segment seems to be regulated by the corresponding UTRs. For example, recombinant BUNV containing a L segment CDS with M segment UTRs showed slower replication rates than the wild type strain [58]. In addition, deletions in the BUNV L and M UTRs also led to slower replication rates [57]. How the extended UTRs of the BKAV M segment regulate transcription and translation has to be studied in more detail.

In this study, WYOV was detected for the first time in Mexico. Neither WYOV nor anti-WYOV antibodies have been found in Mexico so far [4]. Viruses of the Wyeomyia virus serogroup were detected in mosquitoes from Brazil, Colombia, Trinidad, and Peru in the middle and the end of the 20th century [59,60]. Antibodies were found in birds and in healthy humans [59,61,62]. WYOV was isolated from a patient with febrile illness in Panama in 1963 and is believed to cause mild disease [63]. The detection of WYOV in a primary rainforest area in Mexico concomitant with the comparatively high nt and aa distances of 13% and 3.2% within the highly conserved region of the RdRp gene to other WYOV strains suggest that WYOV has been endemic in Central America for a long time and was not recently introduced from South America.

The discovery of LAKV in a single *Wyeomyia complosa* mosquito has greatly expanded our knowledge on the genetic diversity and geographic distribution of insect-associated viruses. LAKV defines a new deep rooting lineage in distant relationship to the insect-specific viruses of the genera *Herbevirus* and *Shangavirus* from Africa and Asia. The LAKV RdRp and N proteins show distances of 74–79% and 69–86% to peribunyaviruses from the established genera suggesting that LAKV belongs to a new genus. Interestingly, LAKV shares genome characteristics with orthobunya- and herbeviruses, providing evidence that it cannot be assigned to one of the established genera. Like herbeviruses, LAKV lacks NSs and NSm proteins, as well as the variable region of the Gc protein, but it contains the conserved genome termini of orthobunyaviruses. The genome organization, genetic distance to other peribunyaviruses, and phylogenetic position of LAKV suggest that it has a host range restriction to insects. Unfortunately, virus isolation attempts in cell culture failed and experimental evidence for a host range restriction to insects is lacking. Virus isolation attempts may have failed due to the high divergence of the LAKV Gc fusion protein to that of peribunya- and herbeviruses. The Gc glycoprotein is involved in virus attachment and pH-mediated membrane fusion and cell entry [3,64]. The loop structure of the Gc fusion protein is conserved and mediated via conserved amino acids [65,66]. However, only very few of these conserved amino acids were present in the LAKV Gc fusion protein. An explanation might be that the LAKV Gc fusion protein is specific for *Wyeomyia complosa* mosquitoes, which belong to the tribe Sabethini, and cannot interact with cells derived from *Aedes albopictus* mosquitoes, which belong to the tribe Aedini. No viruses have been detected in *Wyeomyia complosa* mosquitoes so far.

In summary, we have identified and characterized a great genetic diversity of peribunyaviruses in mosquitoes originating from the area of the Palenque National Park, Mexico. The novel viruses show surprising differences to the characteristics of known peribunyaviruses. Our findings show that surveillance of mosquitoes from remote regions, which include mosquito species that are rarely found, will help to increase our understanding of the immense diversity of mosquito-associated viruses. 

## Figures and Tables

**Figure 1 viruses-11-00832-f001:**
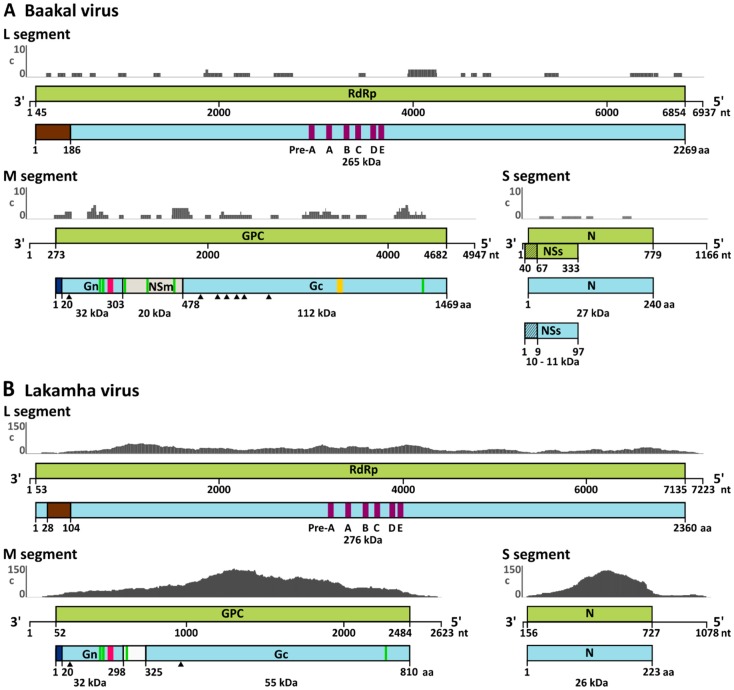
Schematic genome organization of Baakal (BKAV) (**A**) and Lakamha (LAKV) (**B**) viruses. Open reading frames (ORF) are shown in green boxes and predicted corresponding proteins are depicted as light blue boxes. Genome positions and predicted protein masses are indicated. The endonuclease domain is shown as a brown box. The putative signal peptide is indicated by a dark blue box. Predicted N-linked glycosylation sites are shown as black triangles. Green boxes are representing putative transmembrane domains. Gn zinc finger motif and Gc fusion peptide motif are depicted as pink and orange boxes, respectively. The graph above each genome segment shows the distribution and number of obtained NGS reads. The *y*-axis indicates the coverage (c), the maximum number of overlapping reads per genome position.

**Figure 2 viruses-11-00832-f002:**
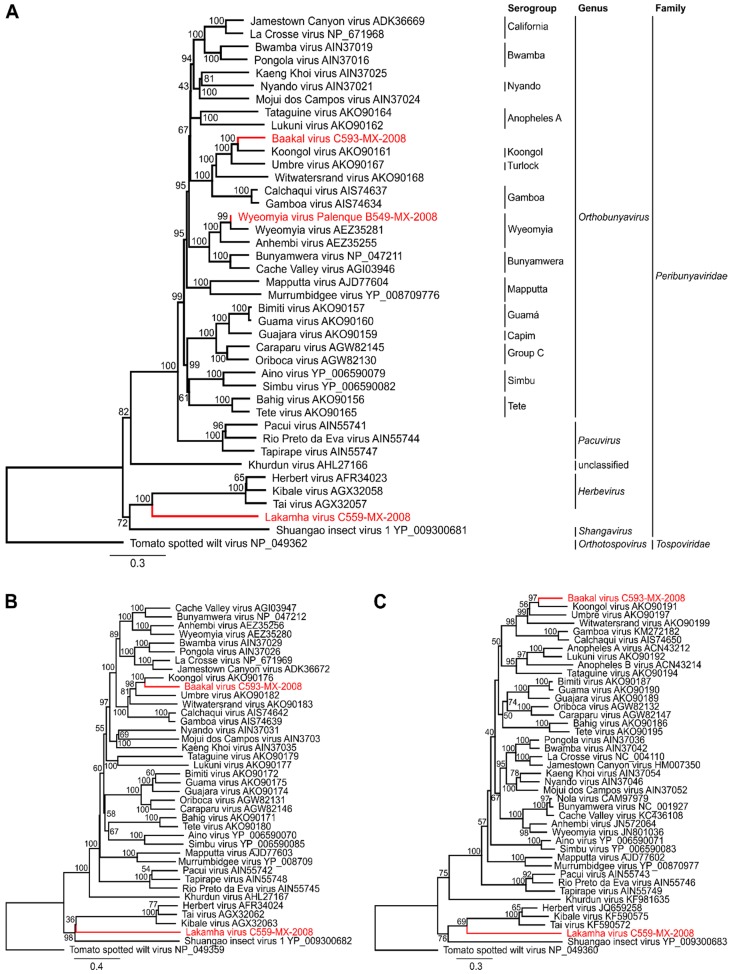
Phylogeny of representative peribunyaviruses and the three novel Mexican viruses. Maximum likelihood [27] trees of the RdRp protein (**A**), the glycoprotein precursor (**B**) and the nucleocapsid (**C**) are shown. For phylogenetic inference the deduced aa sequences of each genome segment were aligned with sequences of representative peribunyaviruses. ML trees were calculated using the Blosum62 substitution model. Tree topology was supported by bootstrap analyses with 1000 replicates. Tomato spotted wilt virus (family *Tospoviridae*, genus *Orthotospovirus*) was used as an outgroup. Accession numbers are depicted behind species names. Viruses identified in this study are shown in red.

**Figure 3 viruses-11-00832-f003:**
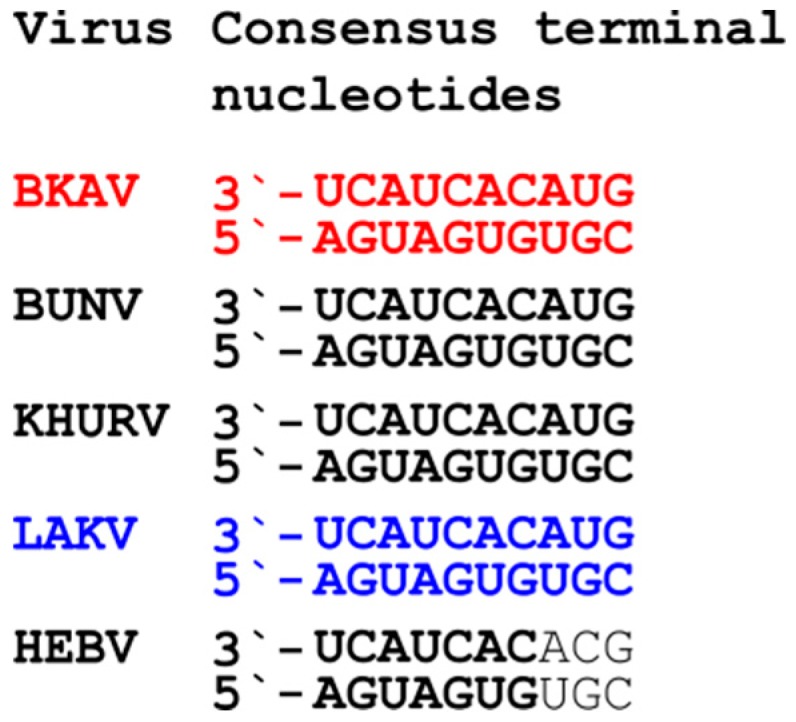
Conserved terminal nucleotides. The genome termini of the BKAV (**red**) and LAKV (**blue**) compared to representative peribunyaviruses are shown. BUNV, Bunyamwerea virus (genus *Orthobunyavirus*); KHURV, Khurdun virus (unclassified); and HEBV, Herbert virus (genus *Herbevirus*).

**Figure 4 viruses-11-00832-f004:**
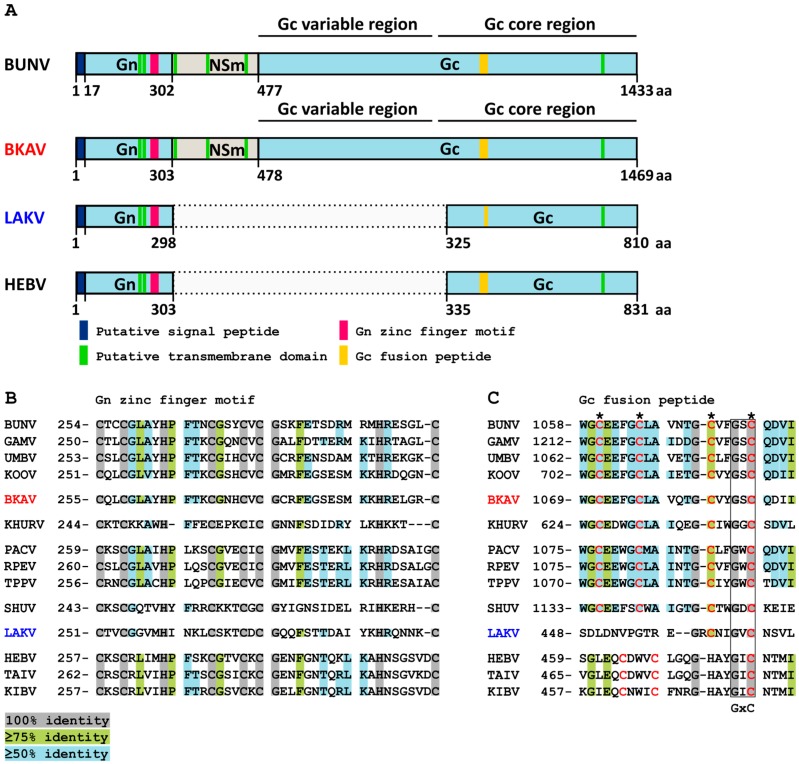
Peribunyavirus glycoprotein precursor protein organization. (**A**) Diagram of glycoprotein precursor protein domains. Regions not present in LAKV and HEBV are indicated as dotted box. Alignment of conserved amino acids of the (**B**) Gn zinc finger motif and the (**C**) Gc fusion petide. Conserved amino acids shaded in grey, light green and light blue indicate 100%, 75%, or 50% identity, respectively. Conserved cysteins are shown in red.

**Figure 5 viruses-11-00832-f005:**
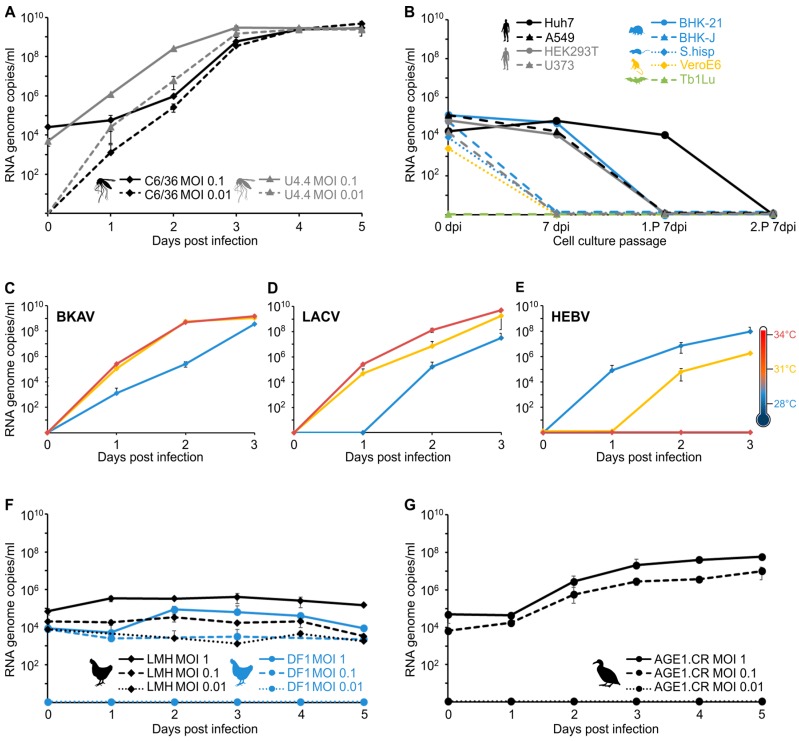
Replication of BKAV in different insect and vertebrate cell lines. (**A**) The insect cells C6/36 and U4.4 were infected in duplicate with MOI 0.1 and 0.01. Samples were taken daily over a five-day period. (**B**) The mammalian cell lines BHK-21, BHK-J, VeroE6, Huh7, HEK293T, A549, U373, Tb1Lu, and S.hisp were infected with MOI 10 and cell culture supernatant was passaged twice onto fresh cells after seven days. Samples were taken on day of infection (0 dpi), after 7 days (7 dpi) and at the end of passage one (1.P 7 dpi) and two (2.P 7 dpi). (**C**) BKAV replication in C6/36 cells was investigated at three different temperatures and compared to that of (**D**) La Crosse virus (LACV) and (**E**) Herbert virus (HEBV). Cells were infected in duplicates with an MOI of 0.01. Samples were taken daily for three days. Further BKAV replication was investigated in the two chicken cell lines LMH and DF1 (**F**) and in the anatine cell line AGE1.CR (**G**). Cells were infected in duplicates with MOIs of 1, 0.1, and 0.01. Samples were taken daily over a five-day period. Viral RNA genome copy numbers in all investigated cell culture supernatants were measured by real-time RT-PCR.

**Table 1 viruses-11-00832-t001:** Viruses identified in this study with additional information on mosquito species and collection site.

Virus	Strain	Mosquito Species (Similarity COI in %)	Collection Site
Baakal virus	C593-MX-2008	*Culex nigripalpus* (99.67%)	Rainforest
Wyeomyia virus Palenque	B549-MX-2008	*Culicinae* sp., Tribe *Sabethini*	Edge
Lakamha virus	C559-MX-2008	*Wyeomyia complosa* (98.17%)	Edge

**Table 2 viruses-11-00832-t002:** Number of BKAV and LAKV reads obtained from deep sequencing.

Virus	Number of Reads/Sample		Segment Size (nt)	Genome Coverage (nt)	Coverage (%)
Baakal virus	Total	1979			
	Virus specific	59			
	L segment	21	6937	1986	28.6
	M segment	34	4947	2322	46.9
	S segment	4	1166	301	25.8
Lakamha virus	Total	1,146,244			
	Virus specific	1289			
	L segment	547	7223	7082	98.0
	M segment	557	2623	2521	96.1
	S segment	185	1078	1053	97.7

**Table 3 viruses-11-00832-t003:** RdRP (bottom left) and nucleocapsid (top right) amino acid pairwise identity values of the Baakal virus (BKAV) and the Lakamha virus (LAKV) compared to selected peribunyavirus species of the genera *Orthobunyavirus*, *Pacuvirus*, *Shangavirus* and *Herbevirus*.

	Orthobunyavirus											Pacuvirus			Uncl.	Shanga-virus	Uncl.	Herbevirus		
	BKAV	KOOV	UMBV	WITV	CALV	GAMV	BUNV	CVV	WYOV	JCV	LACV	PACV	RPEV	TPPV	KHURV	SHUV	LAKV	HEBV	TAIV	KIBV
BKAV		62.82	52.74	49.19	40.76	43.28	33.19	33.62	35.74	40.51	39.24	25.70	26.51	26.21	18.60	18.82	15.94	14.68	16.33	14.68
KOOV	75.46		54.66	46.34	43.16	44.02	31.91	33.62	33.62	38.14	39.83	26.05	27.73	28.99	18.60	18.15	13.55	15.08	17.13	16.67
UMBV	68.27	70.35		44.35	43.04	43.88	35.59	36.44	41.53	37.55	39.24	25.83	25.83	25.42	19.67	18.37	15.42	14.96	17.00	16.93
WITV	57.92	58.69	58.83		35.37	37.40	34.01	34.01	34.82	33.47	35.08	25.60	28.40	28.40	18.11	18.15	14.12	14.45	14.89	14.07
CALV	59.72	59.89	59.73	58.14		85.29	34.89	35.32	34.89	35.86	37.13	26.03	25.21	24.38	20.25	16.49	17.53	15.87	17.13	18.25
GAMV	59.63	60.09	58.54	57.22	93.56		35.74	34.89	35.74	35.86	35.86	26.45	25.21	25.62	19.42	16.84	16.33	14.29	16.33	17.06
BUNV	47.62	47.27	47.01	46.78	49.49	49.05		90.56	63.52	44.44	43.16	32.63	29.66	30.93	18.67	15.66	17.60	18.33	19.20	16.73
CVV	47.18	47.80	47.14	46.65	49.71	49.27	82.09		63.95	43.59	43.59	32.20	29.24	31.78	20.33	17.08	18.00	17.13	18.80	16.33
WYOV	48.28	49.08	47.45	47.57	49.62	49.19	67.83	67.20		47.86	48.72	31.78	28.81	29.66	19.50	18.51	18.40	17.93	17.20	16.33
JCV	50.48	50.92	51.51	50.50	54.44	54.71	55.08	54.46	54.99		82.13	27.31	28.99	25.63	18.60	17.67	17.13	16.27	17.13	14.29
LACV	51.41	51.54	51.55	51.34	54.84	54.57	55.87	55.34	55.08	83.47		29.83	28.99	26.89	18.18	18.73	15.54	15.87	15.14	15.48
PACV	39.96	40.14	40.22	40.18	42.28	41.86	42.80	43.02	43.67	43.53	43.31		64.63	55.10	17.70	15.59	17.46	16.60	15.08	15.42
RPEV	40.18	40.63	40.26	40.48	41.48	40.81	43.68	43.59	44.15	43.36	43.75	69.68		55.92	16.46	15.59	17.06	18.18	16.27	14.62
TPPV	39.29	40.26	40.38	40.97	42.26	42.07	43.50	43.50	43.48	44.26	44.35	69.06	68.09		14.88	15.99	17.93	17.46	15.94	17.46
KHURV	29.13	29.73	29.04	28.50	29.28	28.96	30.90	30.78	31.22	30.20	30.33	30.95	31.30	31.04		12.54	17.36	16.60	15.04	16.60
SHUV	20.63	21.16	21.21	21.38	21.19	21.22	20.58	20.66	20.88	20.74	21.13	20.41	20.12	19.90	21.53		14.91	15.05	13.31	13.62
LAKV	23.69	24.57	24.44	23.98	25.44	25.64	25.74	25.70	25.65	25.52	26.37	25.78	25.87	25.48	28.36	21.12		23.43	22.27	23.43
HEBV	24.57	24.51	23.93	23.74	24.52	24.41	24.91	24.83	24.75	23.97	24.44	24.10	24.10	24.49	27.03	20.20	30.83		66.22	72.57
TAIV	24.31	23.97	24.02	23.75	24.33	24.18	25.04	25.52	25.08	24.10	24.57	23.91	23.87	24.46	27.63	20.16	30.85	79.31		60.89
KIBV	24.43	24.21	24.10	24.06	24.57	24.38	24.68	25.36	24.88	23.90	24.30	23.91	24.35	24.74	27.12	19.77	30.45	80.02	78.61	✕

The established peribunyavirus genera are highlighted as follows: *Orthobunyavirus,* green, *Pacuvirus,* blue, *Shangavirus,* yellow and *Herbevirus,* light grey. The unclassified Khurdun virus (KHURV) and the novel Lakamha virus (LAKV) are highlighted in dark grey and light orange, respectively. Viruses identified in this study are depicted in red.

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
