# Peer review of "Detection of Two Highly Diverse Peribunyaviruses in Mosquitoes from Palenque, Mexico"

_viruses, 2019, doi:10.3390/v11090832_

Round 1

Reviewer 1 Report

Kopp and colleagues describe the identification and characterisation of three peribunyaviruses, detected in mosquitoes collected from Mexico. Of the two new viruses detected, one could not be isolated, but the complete genome was elucidated from RNA extracted from individual mosquitoes. Phylogenetic analysis of this virus (LAKV), indicates that it should likely be assigned to a new genus. The second virus, BKAV showed surprising host specificity. While it replicated to very high titres in mosquito cells, it did not replicate in any of the vertebrate cell lines assessed, apart from those derived from Muscovy ducks. The ability to replicate in vertebrate cells was also in concordance with the virus’s ability to replicate in mosquito cells at higher temperatures.

This is a well-written and interesting collection of work. I look forward to reading of future investigations with BKAV. I have only minor corrections for consideration.

Line 51 – Nucleocapsid probably doesn’t need capitalisation Please check the methods of section 2.1. There are no details on the RT step. Was this a one-step reaction? Methods section 2.4 – please provide extra details on the passaging – was it blind passaging and dilution of the culture supernatant? Throughout the manuscript, there are often full stops where commas should be and vice versa – e.g. line 190 should read 1/3,491, not 1/3.491. This occurs also in Table 1, Table 2, Table 3. In 3.3, please highlight that a putative Nsm was identified in the genome of BKAV. Please clarify the sentence of line 278. BKAV did replicate to higher titres in U4.4 cells at earlier time points, but by day 4, the titres are the same. The sentence of line 291 requires re-phrasing.

Author Response

Kopp and colleagues describe the identification and characterisation of three peribunyaviruses, detected in mosquitoes collected from Mexico. Of the two new viruses detected, one could not be isolated, but the complete genome was elucidated from RNA extracted from individual mosquitoes. Phylogenetic analysis of this virus (LAKV), indicates that it should likely be assigned to a new genus. The second virus, BKAV showed surprising host specificity. While it replicated to very high titres in mosquito cells, it did not replicate in any of the vertebrate cell lines assessed, apart from those derived from Muscovy ducks. The ability to replicate in vertebrate cells was also in concordance with the virus’s ability to replicate in mosquito cells at higher temperatures.

This is a well-written and interesting collection of work. I look forward to reading of future investigations with BKAV. I have only minor corrections for consideration.

Re: Thank you very much for the positive evaluation of our work.

Line 51 – Nucleocapsid probably doesn’t need capitalisation.

Re: Was corrected as suggested.

Please check the methods of section 2.1. There are no details on the RT step. Was this a one-step reaction?

Re: Details on RT step have also been performed as described previously. This information together with information on PCR have now been included in the methods section 2.1. The text now reads:

“Mosquito identification, RNA extraction and cDNA synthesis was performed as described previously [24]. Mosquitoes were tested in pools of ten specimens for peribunyaviruses by a generic reverse transcription (RT) PCR targeting the RdRp gene using Platinum® Taq polymerase (Life Technologies, Darmstadt, Germany). The first round PCR mixture (25 µl) contained 2 µl cDNA as template, 1x buffer, 2.5 mM MgCl2, 0.2 mM dNTPs, 0.6 µM forward and reverse primer, 0.1 µl Platinum Taq polymerase. Components and concentrations of the hemi nested PCR mixture were similar to the mixture described above, but 0.5 µl of the first round PCR product served as template. First round PCR was carried out with the primers Peri-F1 5`‑CAAARAACAGCAAAAGAYAGRGARA and Peri-R1 5`‑TTCAAATTCCCYTGIARCCARTT, followed by a hemi nested PCR with Peri-F2 5`‑ATGATTAGYAGRCCDGGHGA and Peri-R1, respectively. The thermal cycling protocol included the following steps: 3 min 95 °C, ten touch down cycles of 15 s 95 °C, 20 s 55 °C (-0.5 °C per cycle), 40 s 72 °C, 35 cycles of 15 s 95 °C, 20 s 50 °C, 40 s 72 °C and a final elongation step at 72 °C for 5 min.“

Methods section 2.4 – please provide extra details on the passaging – was it blind passaging and dilution of the culture supernatant?

Re: These details are now also provided in the methods section 2.4:

Supernatants of the vertebrate cell lines VeroE6, BHK‑J, BHK-21, Tb1Lu, S.hisp, Huh7, A549, HEK293T and U373 infected with MOI 10 were passaged two additional times after 7 days post infection. Blind cell culture passages with 100 µl cell culture supernatant of the previous cell culture passage mixed with 200 µl media without additives were used for inoculation of fresh cells.“

Throughout the manuscript, there are often full stops where commas should be and vice versa – e.g. line 190 should read 1/3,491, not 1/3.491. This occurs also in Table 1, Table 2, Table 3.

Re: Thank you for this comment. This has been corrected in the entire manuscript.

In 3.3, please highlight that a putative Nsm was identified in the genome of BKAV.

Re: This is now mentioned in the text and the text now reads:

“The full genome of BKAV showed the typical genome organization of orthobunyaviruses including the three segments L, M and S and the conserved terminal nucleotides. The glycoprotein precursor (GPC), encoded on the M segment, showed the conserved protein cleavage sites and is predicted to be cleaved into Gn, NSm and Gc proteins (Figure 1A, Figure 3).“

Please clarify the sentence of line 278. BKAV did replicate to higher titres in U4.4 cells at earlier time points, but by day 4, the titres are the same.

Re: The sentence has been clarified and now reads:

BKAV replicated up to 100-fold better on immune competent mosquito cells (U4.4) than on DICER deficient mosquito cells (C6/36) at 1 and 2 dpi (Figure 5A). However, BKAV reached a similar plateau of >109 RNA genome copies/ml 4 dpi on both cell lines.“

The sentence of line 291 requires re-phrasing.

Re: The sentence has been corrected and now reads:

“BKAV replication was 10- to 100-fold more efficient at temperatures of 31° and 34°C compared to lower temperature of 28°C (Figure 5C) suggesting that BKAV is not an insect-specific virus.“

Reviewer 2 Report

In the manuscript “Detection of two highly diverse peribunyaviruses in mosquitos from Palenque, Mexico,” the authors describe the results of mosquito surveillance for peribunyaviruses in Mexico. The authors describe the first detection of Wyeomyia virus in Mexico, and two previously undescribed peribunyaviruses. The study presented emphasizes the degree of which peribunyaviruses are under recognized and adds to the knowledge of peribunyavirus genetic plasticity by description of a potential new genus. A few minor suggestions for improvement follow.

Despite the significant abundance of LAKV reads (Table 2), the authors failed to isolate a virus. The discussion of the low identity of the Gc fusion motif (Lines410-412, Figure 4) are suggested as reason for lack of isolation. Can the authors elaborate on this theory? For example, is there a significant structural change? The very low identity of LAKV Gc to other peribunyavirus Gc is striking. It would be suggested to confirm this region with independent sequencing methods such as Sanger sequencing.

In lines 243-245, the authors describe the UTR of BAKV to be “much longer than other orthobunyaviruses”. Please be more specific, how much longer? It would be suggested to directly compare BAKV UTR to koongal and/or Umbre UTRs.

It is suggested to shorten the introduction to focus on the importance of the surveillance and emerging viruses.

Author Response

In the manuscript “Detection of two highly diverse peribunyaviruses in mosquitos from Palenque, Mexico,” the authors describe the results of mosquito surveillance for peribunyaviruses in Mexico. The authors describe the first detection of Wyeomyia virus in Mexico, and two previously undescribed peribunyaviruses. The study presented emphasizes the degree of which peribunyaviruses are under recognized and adds to the knowledge of peribunyavirus genetic plasticity by description of a potential new genus. A few minor suggestions for improvement follow.

Despite the significant abundance of LAKV reads (Table 2), the authors failed to isolate a virus. The discussion of the low identity of the Gc fusion motif (Lines 410-412, Figure 4) are suggested as reason for lack of isolation. Can the authors elaborate on this theory? For example, is there a significant structural change? The very low identity of LAKV Gc to other peribunyavirus Gc is striking. It would be suggested to confirm this region with independent sequencing methods such as Sanger sequencing.

Re: Thank you for your comments on the lack of isolation of LAKV and on the divergence of the LAKV Gc protein to that of other peribunyaviruses. As suggested, we now provide more detail on this topic including a novel figure (Fig. 4A). The region of interest is covered by 50 – 150 reads per genome position of the Gc protein. There has been not a single ambigious position, nor any region. We agree that it can be difficult to correctly assembly genes with great divergence to known sequences. However, due to the sequence depth and high coverage of reads, this is not the case here. We thus are absolutely sure that the NGS derived sequence of the Gc gene is correct. The text now reads:

Virus isolation attempts may have failed due to the high divergence of the LAKV Gc fusion protein to that of peribunya- and herbeviruses. The Gc glycoprotein is involved in virus attachment and pH-mediated membrane fusion and cell entry [3, 64]. The loop structure of the Gc fusion protein is conserved and mediated via conserved amino acids [65, 66]. However, only very few of these conserved amino acids were present in the LAKV Gc fusion protein. An explanation might be that the LAKV Gc fusion protein is specific for Wyeomyia complosa mosquitoes, which belong to the tribe Sabethini, and cannot interact with cells derived from Aedes albopictus mosquitoes, which belong to the tribe Aedini. No viruses have been detected in Wyeomyia complosa mosquitoes so far.”

In lines 243-245, the authors describe the UTR of BAKV to be “much longer than other orthobunyaviruses”. Please be more specific, how much longer? It would be suggested to directly compare BAKV UTR to koongal and/or Umbre UTRs.

Re: This information is now included in the text and the text now reads:

“Interestingly, the 3’ and 5’ UTRs of the BKAV M segment are of similar length (272 nt and 265 nt, respectively) and are much longer than the UTRs of other orthobunyaviruses (e.g. BUNV 3’UTR is 56 nt and the 5’ UTR is 100 nt ). Unfortunately, no information on KOOV and UMBV UTRs is available.“

It is suggested to shorten the introduction to focus on the importance of the surveillance and emerging viruses.

Re: Thank you for this comment and for the suggestion to focus on the importance of surveillance and emerging viruses. An entire paragraph (lines 73 – 85) is dedicated to this topic. We decided not to shorten the rest of the introduction as these parts give an overview on the current knowledge on peribunyaviruses and are required to understand the results of the study.